# Association between Successful Palpation of the Cricothyroid Membrane and the 3-3-2 Rule for Predicting Difficult Airway in Female Patients Undergoing Non-Neck Surgery: A Prospective Observational Cohort Study

**DOI:** 10.3390/jcm11092316

**Published:** 2022-04-21

**Authors:** So Yeon Lee, Da Kyung Hong, Chang Jae Kim, Mee Young Chung, Sanghoon Lee, Min Suk Chae

**Affiliations:** 1Department of Anesthesiology and Pain Medicine, Eunpyeong St. Mary’s Hospital, College of Medicine, The Catholic University of Korea, Seoul 06591, Korea; soyeon0719@naver.com (S.Y.L.); hongdk1022@naver.com (D.K.H.); ksw070591@catholic.ac.kr (C.J.K.); jhjs0806@catholic.ac.kr (M.Y.C.); 2Department of Anesthesiology and Pain Medicine, Yeouido St. Mary’s Hospital, College of Medicine, The Catholic University of Korea, Seoul 06591, Korea; sah1219@naver.com; 3Department of Anesthesiology and Pain Medicine, Seoul St. Mary’s Hospital, College of Medicine, The Catholic University of Korea, 222, Banpo-daero, Seocho-gu, Seoul 06591, Korea

**Keywords:** difficult airway, 3-3-2 rule, cricothyroidotmy, cricothyroid membrane, POCUS

## Abstract

Background: Prediction of difficult airway is important for airway management in patients undergoing surgery. The assessment of airway structures and establishment of protective airway strategies are essential to improve patient safety. However, the association between successful palpation of the cricothyroid membrane and airway predictions has not been fully elucidated in patients undergoing surgery. We investigated this in female patients undergoing non-neck surgery. Methods: A total of 68 female patients were enrolled in this prospective observational cohort study between January 2021 and June 2021 at Eunpyeong St. Mary’s Hospital, College of Medicine, Catholic University of Korea, Seoul, Republic of Korea. Exclusion criteria were male patients and those with neck pathology or neck surgery. The assessment of difficult airway was performed before the induction of anesthesia and was defined by one of the following conditions: inter-incisor distance < 3 fingerbreadths, hyoid-to-mental distance < 3 fingerbreadths, and thyroid-to-hyoid distance < 2 fingerbreadths (the “3-3-2 rule”). The accuracy of palpable identification of the cricothyroid membrane was confirmed by ultrasonography (US). The patients were divided into the non-difficult airway (NDA) group (*n* = 30) and the difficult airway (DA) group (*n* = 30). Results: The two groups were comparable in terms of age, but the DA group had higher body mass index (BMI). In airway assessment, 9 patients showed inter-incisor distance < 3 fingerbreadths, 3 patients showed hyoid-to-mental distance < 3 fingerbreadths, and 24 patients showed thyroid-to-hyoid distance < 2 fingerbreadths in the DA group. The rate of successful palpation of the cricothyroid membrane was higher in the patients without than in those with difficult airway variables. Conclusions: Patients with a positive 3-3-2 rule showed a poor palpability of cricothyroid membrane.

## 1. Introduction

The prediction of a difficult airway is crucial for successful intubation and can minimize both physical and psychological trauma to patients [1]. Patients with difficult airways may have higher rates of morbidity and mortality related to adverse airway events during the perioperative period [2,3]. Cricothyroidotomy is an emergency method to maintain airway patency under life-threatening conditions, such as “Can’t intubate, can’t ventilate (CICV)” or “Can’t intubate, can’t oxygenate (CICO).” The success of this rescue technique relies on the accurate and rapid identification of an anatomical puncture landmark, i.e., the cricothyroid membrane (CTM). To identify the CTM appropriately, clinicians must be familiar with the relevant anatomical structures that represent the boundaries of the CTM, i.e., the thyroid cartilage superiorly, the cricoid cartilage inferiorly, and the cricothyroideus muscle laterally on both sides. However, palpation of the CTM may lead to the misidentification and misdiagnosis of the airway structure. In a previous study, Elliott et al. reported that only 30% of anesthetists accurately identified the skin over the CTM with landmark palpation [4]. Furthermore, the identification of the CTM is more difficult in women than in men because of the less prominent laryngeal prominence. To avoid these difficulties, bedside ultrasonography (US) has been proposed as a method for identifying the CTM.

A number of different tools have been suggested for screening, predicting, and managing difficult airway, including the Mallampati score; LEMON score; body mass index (BMI); neck extension or tooth morphology; mouth opening, oropharyngeal classification, thyromental distance, sternomental distance, and Wilson risk score; and the “3-3-2 rule” [5,6,7,8,9,10,11,12,13]. The 3-3-2 rule is simple and immediately available to measure the airway according to three spatial assessments in the operating room: interincisor distance < 3 fingerbreadths, hyoid-to-mental distance < 3 fingerbreadths, and thyroid-to-hyoid distance < 2 fingerbreadths [14]. Each measurement is an important determinant of successful direct laryngoscopy, and the risk of failure of the airway approach increases if any individual factor is inadequate [7].

However, the role of a difficult airway screening tool for identifying the CTM, as the first airway rescue step, has not been fully investigated in the operating room setting. There is a paucity of literature regarding the use of digital palpation (DP) and US for CTM identification prior to cricothyrotomy. Therefore, we investigated the effects of a difficult airway as determined by the 3-3-2 rule on the accuracy of CTM palpation in female patients undergoing non-neck surgery. We examined the airway before induction of anesthesia and performed a comparative analysis of the success rates of CTM identification between difficult airway (DA) and non-difficult airway (NDA) groups.

## 2. Materials and Methods

### 2.1. Ethical Considerations

This prospective observational study was approved by the Institutional Review Board of Eunpyeong St. Mary’s Hospital Ethics Committee, College of Medicine, Catholic University of Korea, Seoul, Republic of Korea (Approval number: PC21OISI0014), on 28 January 2021, and was performed in accordance with the principles of the Declaration of Helsinki. The trial was registered in the Clinical Research Information Service (CRIS), Republic of Korea (registration number: KCT0005902), on 17 February 2021. A total of 60 female patients with American Society of Anesthesiologists (ASA) physical status I or II undergoing surgery under general or regional anesthesia between January 2021 and June 2021 were enrolled in the study. All participants provided written informed consent prior to enrollment.

### 2.2. Study Population

A total of 68 adult (≥19 years) female patients with ASA physical status I or II scheduled to undergo elective non-neck surgery were enrolled. Exclusion criteria were pregnancy, known neck pathology (*n* = 3), history of neck surgery (*n* = 2), refusal to participate in the study (*n* = 3), inability to understand the study due to cognitive impairments, and ASA physical status ≥ III. Ultimately, 60 patients were included (Figure 1).

In the operating room, attending certificated anesthesiologists who assessed the airway using the 3-3-2 rule and CTM palpation were aware of the group assignments but did not participate in the study or data collection.

#### 3-3-2 Rule for Predicting Difficult Airway

Attending certificated anesthesiologists examined the airway immediately before the induction of anesthesia based on three factors (i.e., the 3-3-2 rule) and a difficult airway was defined by the presence of one or more of the variables: the distance between the upper and lower incisors, the distance between the hyoid bone and the chin, and the distance from the thyroid cartilage to the floor of the mouth (Figure 2).

On the operating table, the patients lay on a pillow in the sniffing position, which is considered the optimal head position for direct laryngoscopy [15]. The patients were asked to open their mouth as far possible to determine whether it was possible to admit the researcher’s three fingers. Then, we determined whether the distance between the mentum and the hyoid bone was greater than three fingerbreadths, and whether the space between the superior notch of the hyoid bone and the thyroid cartilage was less than or greater than two fingers.

Patients were divided into two groups based on physical examination (palpation) as the NDA group in which one or more of the 3-3-2 rules was applicable and the DA group in which none of the 3-3-2 rules was applicable.

### 2.3. Palpation and Confirmation of the Cricothyroid Membrane

After 3-3-2 rule assessment, the attending anesthesiologists directly palpated the patient’s neck and identified and marked the CTM by pressing with the fingernail (Figure 3). CTM palpation for identification of the CTM was performed using the index and third fingers of the nondominant hand to palpate the thyroid cartilage at the midline starting from cephalad (near the hyoid bone) and moving caudally to the cricothyroid cartilage.

Then, three authors (SYL, CJK, and MYC) individually assessed whether CTM palpation was successful using bedside US (LOGIQ™ E10 Ultrasound; GE Healthcare, Waukesha, WI, USA) with a linear high-frequency transducer (8–12 MHz; Figure 4 and Figure 5).

Identification of the CTM was considered correct if the mark was between the superior and inferior borders and within 0.5 cm of the midline [16]. Successful CTM palpation and marking was defined as when the CTM was visible immediately on the US monitor or when two or more authors confirmed the accuracy of surface landmark palpation for CTM identification.

### 2.4. Clinical Variables

The demographic characteristics, such as age, height, weight, and BMI, were compared between the two groups. As the main outcome, the success rate of digital CTM palpation was analyzed between the groups.

### 2.5. Sample Size and Statistical Methods

Preliminary analyses in a pilot study conducted prior to the full-scale study suggested that the NDA and DA groups would have successful CTM palpation rates of 90% and 55%, respectively. With a risk for type 1 error of 5%, risk for type 2 error of 20%, and dropout rate of 10%, 30 patients were required in each group. Therefore, a total of 60 patients were included in the study.

The normality of the continuous data was assessed using the Shapiro–Wilk test. Categorical variables are reported as frequencies (%), and continuous variables are shown as the mean (standard deviation). The *χ*^2^ test or Fisher’s exact test was used to compare categorical variables, as appropriate. Student’s *t* test was used to compare continuous variables. The predictive accuracy of the 3-3-2 rule for CTM palpation was evaluated with the area under the receiver operating characteristic (ROC) curve (AUC). All statistical tests were two-sided, and differences with *p* < 0.05 were taken to indicate statistical significance. Analyses were performed using SPSS Statistics software (version 22.0; IBM Corp., Armonk, NY, USA) and MedCalc for Windows (V.11.0; MedCalc, Ostend, Belgium).

## 3. Results

There was no significant difference in age distribution between the two groups (*p* = 0.36; Table 1). BMI was significantly higher in the DA group than in the NDA group (*p* = 0.02; Table 1).

In airway assessment of the DA group, 9 patients (30%) showed an interincisor distance < 3 fingerbreadths, 3 patients (10%) showed hyoid-to-mental distance < 3 fingerbreadths, and 24 patients (80%) showed thyroid-to-hyoid distance < 2 fingerbreadths.

The success rate of CTM palpation was significantly higher in the in the NDA group (28/30 = 93%) than in the DA group (21/30 = 70%) (*p* = 0.039; Table 2; Figure 6). As shown in Table 2, the thyroid-to-hyoid distance was a major contributor to the difference between the two groups. The 3-3-2 rule showed a sensitivity of 81.82% (95% CI = 48.2–97.7%) and specificity of 57.14% (95% CI = 42.2–71.2%) for the detection of difficult CTM palpation. We also found that patients in the DA group had a difficult airway during tracheal intubation (*p* = 0.047; Table 2).

## 4. Discussion

Patients with shorter interincisor (<3 fingerbreadths), hyoid-to-mental (<3 fingerbreadths), or thyroid-to-hyoid (<2 fingerbreadths) distances may have higher failure risk for accurate CTM palpation than those with longer distances. In addition, patients with difficult airway variables had a higher BMI than those without these variables despite the comparable age distribution between the two groups.

Although CTM is conventionally identified by palpation between the cricoid cartilage and thyroid cartilage, the classic landmark palpation technique is inaccurate [17]. Other studies related to CTM identification suggested that the DP method may have limitations for accurate and uniform identification of the CTM [18,19,20,21,22,23]. CTM misidentification through DP may be affected by various factors, including female sex, pregnancy, obesity, facial trauma, and other neck pathologies [24,25]. In contrast, Elliot et al. [4] reported that delineation of a cricothyrotomy entry point was not significantly affected by the patient’s weight, height, BMI, neck circumference, or CTM dimensions. You-Ten et al. [24] reported that DP by anesthesiologists was accurate for localizing the CTM in 71% (20/28) of nonobese patients compared to only 39% (11/28) of obese patients (*p* = 0.03). In addition, Aslani et al. [26] reported that the CTM was identified in 24% (10/41) of nonobese versus 0% (0/15) of obese female patients in the neutral neck position (*p* = 0.048). These data were correlated with difficult airway predictor tools that incorporate anatomical characteristics, including the prominence of the thyroid cartilage, the vertical height and width of the CTM, and the anterior neck thickness [27,28]. These findings were consistent with our results that the 3-3-2 rule factors (i.e., interincisor distance, hyoid-to-mental distance, and thyroid-to-hyoid distance) affected the accuracy of successful CTM palpation. Particularly, as a thyroid-to-hyoid distance < 2 fingerbreadths was an indicator of a difficult airway in a large proportion of cases, this distance related to upper airway structure may increase the inability to detect the CTM precisely by palpation. In the emergency room, the cricothyroidotomy procedure under CICO conditions has failure rates as high as 60–75%, exposing patients to high risks of mortality and morbidity [29]. Therefore, more reliable methods for CTM identification are needed. Our results confirm the reproducibility and validity of CTM palpation.

In our study, the success rate of CTM DP was fairly high at 70–90%. This was likely because, to avoid bias according to the degree of training, only trained and experienced anesthesiologists participated in the study, and we excluded residents under training. In a previous prospective observational study conducted at a tertiary emergency department in Saudi Arabia, clinical localization of the CTM using the DP method by emergency medicine trainees was as poor as 30–33% [30]. According to another non-randomized control trial (RCT) in which anesthesiologists evaluated the anterior neck anatomy to identify the CTM, successful identification of the CTM occurred in 15/45 (33%) cases by the conventional palpation method [31]. Experienced anesthesiologists will have much higher degrees of DP skill.

US is an emerging tool to assess airway patency. Rai et al. [32] suggested that US has competitive CTM identification times in subjects with difficult airway anatomy as well as greater accuracy for CTM identification than DP, as it can objectively define neck anatomy to better guide cricothyrotomy and training. The guidelines of the Difficult Airway Society (DAS) recommend US for verification of incision sites if the machine is immediately available and switched on [32]. The utilization of US to improve accuracy and success rates in cases with difficult airway anatomy (e.g., obese and/or pregnant patients) was further supported by the report of You-Ten et al. [24], who showed significantly higher rates of successful CTM identification with US compared to conventional DP in obese patients. A similar trend was seen in patients with abnormal neck anatomy in whom, compared to DP, US had significantly greater accuracy, defined as a distance from target ≤5 mm, and a significantly higher success rate of CTM identification [33]. Multiple studies have shown low accuracy and success rate ≤50% of CTM identification using conventional DP regardless of training level and airway technical skills [4,26,30,34].

As the 3-3-2 rule provides anatomical information on the airway, it may be useful to predict potential difficulty in endotracheal intubation. To avoid airway mismanagement due to poor identification of the CTM, clinicians must be proficient in the use of US, rather than relying solely on palpation. The utilization of US to identify the CTM in female patients who are anticipated to have difficult airway conditions may be beneficial for the progression to cricothyroidotomy. In addition, our results indicate favorable reproducibility using the quick DP method with the 3-3-2 rule and bedside portable US. Our results show that the 3-3-2 rule is simple and has the advantage that similar results can be obtained by other clinicians.

This study has several limitations. As the 3-3-2 rule involves measurement with the fingers, it is difficult to determine precise cutoff values to reduce interobserver variability. In addition, the study population included only female patients without a neck pathology or neck surgery. Therefore, care is required in interpreting and generalizing the findings of this study [35]. As the anesthesiologist assessed the patient with the 3-3-2 rule and already knew about the status (DA or NDA), this may have influenced the estimation of direct palpation.

The 3-3-2 rule was not developed solely for CTM palpation but for the prediction of difficult intubation. There has been insufficient research on predicting the success of its application to the CTM. However, there is some similarity between the factors predicting failed airway rescue care, such as cricothyroidotomy, and the constituents of the 3-3-2 rule. The 3-3-2 rule may be an available tool to screen for difficult CTM palpation in clinical settings. As CTM palpation is the first step in airway rescue, additional studies of the relations between predictive tools for a difficult airway, such as mouth opening, head and neck movement, modified Mallampati, Look-Evaluate-Mallampati-Obstruction-Neck score (LEMON), and CTM palpation, and success or failure or airway rescue are needed.

## Figures and Tables

**Figure 1 jcm-11-02316-f001:**
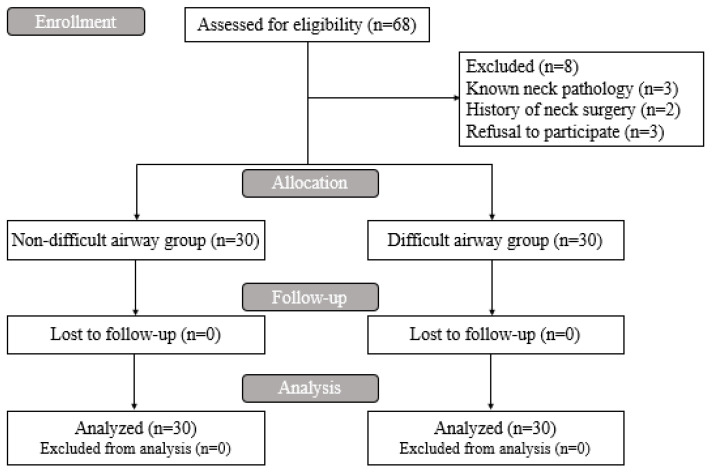
CONSORT flowchart of this clinical study.

**Figure 2 jcm-11-02316-f002:**
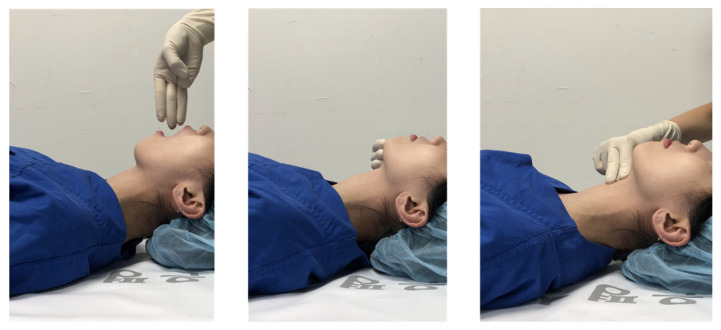
The “3-3-2 rule” for assessing the airway before induction of anesthesia.

**Figure 3 jcm-11-02316-f003:**
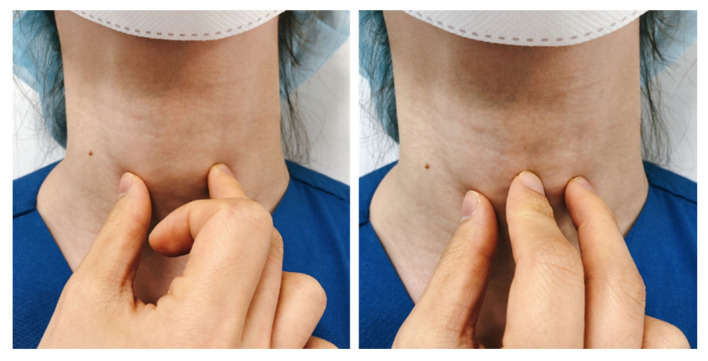
Identifying the cricothyroid membrane by digital palpation.

**Figure 4 jcm-11-02316-f004:**
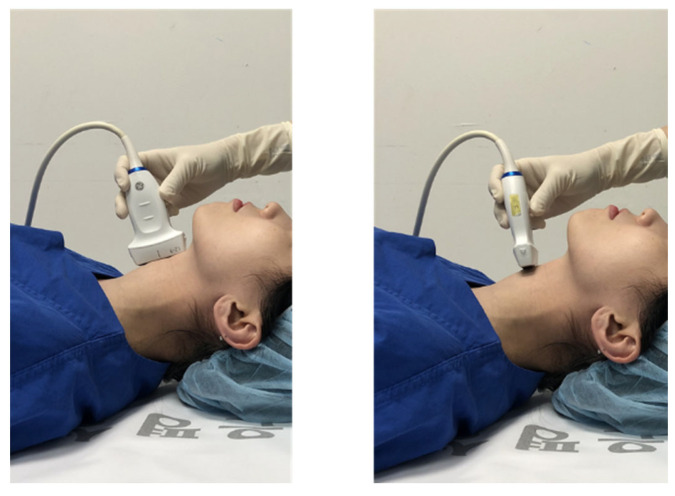
Confirmation of cricothyroid location using bedside ultrasound.

**Figure 5 jcm-11-02316-f005:**
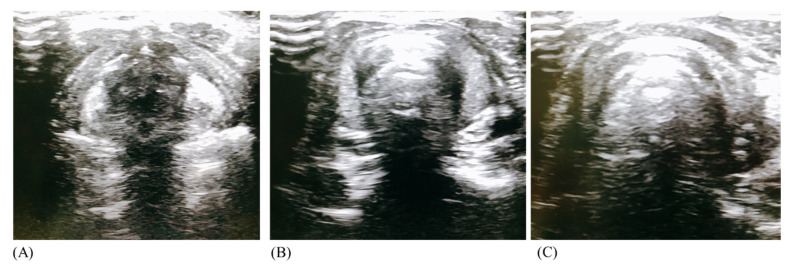
Sonographic appearance of (**A**) Thyroid cartilage (**B**) Cricothyroid membrane (**C**) cricoid cartilage.

**Figure 6 jcm-11-02316-f006:**
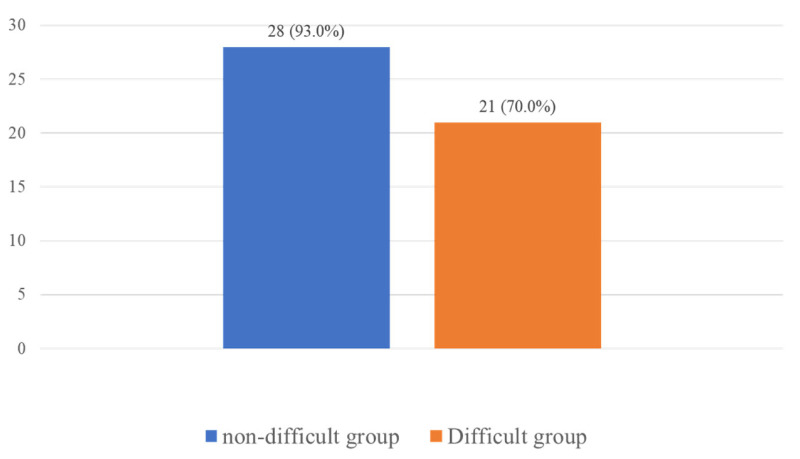
Success rate of cricothyroid membrane palpation.

**Table 1 jcm-11-02316-t001:** Demographic data and airway management.

	Non-Difficult Airway Group	Difficult Airway Group	*p*-Value
*n*	30	30	
Demographic factors			
Age (years)	52.13 ± 14.62	55.77 ± 15.18	0.36
Body mass index (kg/m^2)^	25.15 ± 3.45	27.70 ± 4.67	0.02
Obesity (body mass index ≥30 kg/m^2^)	2 (6.7%)	9 (33.3%)	0.01
**Airway assessment**			
Inter-incisor distance < 3 fingerbreadths	0 (0.0%)	9 (30.0%)	0.002
Hyoid-to-mental distance < 3 fingerbreadths	0 (0.0%)	3 (10.0%)	0.151
Thyroid-to-hyoid distance < 2 fingerbreadths	0 (0.0%)	24 (80.0%)	<0.001

Values are expressed as the mean ± standard deviation or *n* (%).

**Table 2 jcm-11-02316-t002:** Accuracy of cricothyroid membrane palpation according to the 3-3-2 rule for predicting difficult airway in female patients undergoing non-neck surgery.

	Non-Difficult Airway Group	Difficult Airway Group	*p*-Value
*n*	30	30	
Accuracy outcome			
Success rate of cricothyroid membrane palpation, *n* (%)	28 (93.0%)	21 (70.0%)	0.039

*n*: number of subjects.

## Data Availability

The datasets generated and/or analyzed during the current study are not publicly available because disclosing patients’ personal information is against the law, but de-identified datasets are available from the corresponding author on reasonable request.

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
