# Peer review of "Association between Successful Palpation of the Cricothyroid Membrane and the 3-3-2 Rule for Predicting Difficult Airway in Female Patients Undergoing Non-Neck Surgery: A Prospective Observational Cohort Study"

_jcm, 2022, doi:10.3390/jcm11092316_

Round 1

Reviewer 1 Report

Thanks to the Authors to submit their work to JCM. I read it with interest and appreciate the topic of the paper that is interesting and worth of investigation.

The paper presented by the author deals with the identification of the cricothyroid membrane by direct palpation in female patients classified as “difficult airway” (DA) or “non-difficult airway” (NDA). Patients were classified as DA or NDA by the 3-3-2-rule, a method using spatial relationships as determinants of successful direct laryngoscopy.

However, as here commented, some points of the study should be revised to improve the quality and the readability of the paper.

Major Considerations:
As to my knowledge, there is no single indicator in predicting a difficult airway. Even with a combined approach of several methods (mouth opening, upper lip bite tests, thyromental distance, sternomental distance, modified Mallampati and head and neck movement, successful prediction of difficult tracheal intubation is not able to stratify the risk of intubation difficulty. The hyoid-thyroid cartilage distance typically is less than two fingers for many patients; however, the majority of patients do not show a difficult airway during intubation. Please refer to that in the introduction and the discussion.

Moreover you did not report, if the patients you classified by the 3-3-2-rule as “difficult airway” really showed a difficult airway during tracheal intubation.

This limits your conclusion, which I would rephrase to ”patients with a positive 3-3-2-rule show a poor palpability of the cricothyroid membrane”.

It is already known, that some published RCTs have shown the effectiveness of the ultrasound approach for improving the success rate of cricothyroid membrane identification compared with the conventional palpation technique

I suggest to the authors to better explain the reason behind the study and the rationale. Why using solely the 3-3-2-rule? In other terms, which is the novelty of this study and possible implications in the clinical field? Why is that relevant and what gap in the literature is that really addressing?

I am not sure if I understood what the authors wanted to tell: should ultrasonography be performed only in patients with a positive 3-3-2-rule? In my opinion identification of the cricothyroid membrane should be done during the preoperative evaluation whenever possible and that examination should be performed with ultrasound if landmarks are not easy clear.

Introduction

LL 64: I do not understand why you mention the Cormack-Lehane classification. It is not a predictive method.

Methods

LL 104: Why did you not blind the anesthesia providers to the group assignment? If a provider assessed the patient with the 3-3-2-rule he already knows about the status DA or NDA. This may influence the estimation of direct palpation. Please mention that as a limitation.

LL106: How where the patients positioned? Was the neck hyperextended? Was there a pillow under the head? Was the position standardized?

LL107: It is not clear from the manuscript, if the 3-3-2-rule is applicable if only one or all criteria has to be fulfilled. Please specify that more clearly.

LL 124: How was the CTM marked by the provider? By a pen? By pressing finger nail?

LL138: How was success defined? Was there a tolerable distance from the ultrasound assessment?

LL145: What type of preliminary analysis you mean? Is that a published article? Please explain, or add additional data.

Discussion

The discussion section is rather lengthy. Especially the section dealing with pediatric results may be shortened.

Some results are not discussed. I would be interested how you explain a quite high success rate of CTM direct palpation of 70-90%, compared to the data you refer to.

LL 206: Ultrasound is not a tool for improving airway patency; please correct.

LL 206 – 231 refers to ultrasound in comparison to direct palpation. In my opinion accuracy or timing of ultrasound is not the main topic of the study. Please discuss your excellent direct palpation results. Moreover I would be interested in a discussion of the 3-3-2-rule  predictive value for identification of the CTM.

LL 238: The 3-3-2-rule is not a rescue airway step. Please rephrase.

LL 243: What do you mean by “results indicate reproducibility”?

Reviewer 2 Report

  1. The authors basic aim is to test the identification of CTM by palpation vs. USG. Studies have shown that digital palpation is not accurate in children, female gender and obese individuals. The authors wish to test the hypothesisin difficult vs non difficult airway.
  2. Identification of a difficult airway prior to anaesthesia is desirable for an anaesthesiologist. However, even a combination of scores and parameters have failed to provide good efficacy. The use of 3-3-3 rule for identifying a difficult airway is too simplistic a technique. What is the sensitivity or specificity for this technique to detect a difficult airway?
  3. What was the reasons for lower incidence of CTM palpation in the DA group?
  4. The DA group with higher incidence of non-palpation of CTM may be due to higher BMI in that group and not due to difficult airway factors.

Round 2

Reviewer 1 Report

I am sorry but the authors have not implemented my suggestions to improve the manuscript.

The authors were asked to explain the reason behind the study and the rationale. What is the novelty of this study and what are possible implications in the clinical field? Why is that relevant and what gap in the literature is the study really addressing?

None of these questions are answered

Author Response

Reviewer #1

I am sorry but the authors have not implemented my suggestions to improve the manuscript.

The authors were asked to explain the reason behind the study and the rationale. What is the novelty of this study and what are possible implications in the clinical field? Why is that relevant and what gap in the literature is the study really addressing?

None of these questions are answered

Response: We are sorry that our previous responses did not match your queries satisfactory. We hope these responses meet your queries fully.

The establishment of an artificial airway is critical for rescuing critically ill patients. However, it is challenging for anesthesiologists, emergency physicians and critical care physicians to predict a difficult airway early. Successfully predicting a difficult airway enables physicians to offer better treatment options and reduce potential injury. Various methods of difficult airway assessment, including mouth opening degrees, upper lip bite tests, thyromental distance, sternomental distance, modified Mallampati tests, and head and neck movement, are available. To date, single indicator evaluation plays a relatively limited role in predicting a difficult airway. The 3-3-2 rule is used in conjunction with mouth opening (interincisor distance, IID), hyoid-mental distance (HMD) and hyoid-thyroid cartilage distance (HTD) to assess difficult airways. An IID less than three fingers, an HMD less than three fingers, or an HTD less than two fingers may be indicative of a difficult airway. Previous studies suggested that the 3-3-2 rule plays an important role in difficult airway assessment.

In rare life-threatening airway crises of “cannot intubate, cannot oxygenate,” an emergency cricothyrotomy with the insertion of a breathing tube via the cricothyroid membrane is the only option. When performing this potentially life-saving procedure, the first critical step is to palpate and correctly identify the cricothyroid membrane because its misidentification is a major cause of tube misplacements, leading to cricothyrotomy failures and serious complications. However, accurate localization of the cricothyroid membrane using the conventional approach of external palpation is more challenging than anticipated where anesthesiologists, emergency medicine physicians, and trauma surgeons poorly localized the cricothyroid membrane (CTM).

Although the 3-3-2 rule was not developed for predicting difficult CTM identification, the 3-3-2 rule is useful to reflect patency of the upper airway structures and functional motion range consisting of three parts, i.e., mouth opening sufficiency, distance between the mentum and neck/mandible junction (near the hyoid bone), and space between the superior notch of the thyroid cartilage and the neck/mandible junction (near the hyoid bone). These anatomical areas of the 3-3-2 rule share the CTM palpation route where identification of the CTM is performed using the index and third fingers of the nondominant hand to palpate the thyroid cartilage in the midline starting from cephalad (near the hyoid bone) and moving caudally to the cricothyroid cartilage. Therefore, we chose the 3-3-2 rule among the various front of neck (upper airway)-based airway assessments.

To the best of our knowledge, our study is the first to analyze that the predictive accuracy of the 3-3-2 rule for CTM palpation was evaluated based on the area under the receiver operating characteristic (ROC) curve (AUC) (lines 161-162, 165 in page 8), and suggested that the 3-3-2 rule had a sensitivity of 81.82% (95% CI = 48.2–97.7%) and specificity of 57.14% (95% CI = 42.2–71.2%) for detecting difficult CTM palpation (lines 180-181 in page 9). Patients with shorter interincisor (<3 fingerbreadths), hyoid-to-mental (<3 fingerbreadths), or thyroid-to-hyoid (<2 fingerbreadths) distances may have higher failure risk for accurate CTM palpation than those with longer distances (lines 191-192 in page 10). Eventually, the 3-3-2 rule may be an available tool to screen for difficult CTM palpation in clinical settings (lines 261-262 in page 13).  

We hope that the revised manuscript is now suitable for publication in the JCM Special Issue “Airway Management - State of Art.

Thank you for your time.

Reviewer 2 Report

All the queries have been adequately answered.
